# A novel flexible near-infrared endoscopic device that enables real-time artificial intelligence fluorescence tissue characterization

Gareth Gallagher[1], Ra'ed Malallah[1,2], Jonathan P. Epperlein[3], Jeffrey Dalli[1], Niall Hardy[1], Abhinav Jindal[1], Pol G. MacAonghusa[3], Ronan A. Cahill[1]*

1 School of Medicine, The Mater Misericordiae University Hospital and University College Dublin, Dublin, Ireland, 2 Physics Department, Faculty of Science, University of Basrah, Garmat Ali, Basrah, Iraq 3 IBM Research Europe, Dublin, Ireland

* ronan.cahill@ucd.ie

## Abstract

Real-time endoscopic rectal lesion characterization employing artificial intelligence (AI) and near-infrared (NIR) imaging of the fluorescence perfusion indicator agent Indocyanine Green (ICG) has demonstrated promise. However, commercially available fluorescence endoscopes do not possess the flexibility and anatomical reach capabilities of colonoscopy while commercial flexible scopes do not yet provide beyond visible spectral imaging. This limits the application of this AI-NIR classification technology. Here, to close this technical gap, we present our development of a colonoscope-compatible flexible imaging probe for NIR-ICG visualization combined with a full field of view machine learning (ML) algorithm for fluorescence quantification and perfusion pattern cross-correlation (including first in human testing). The imaging probe is capable of 133μm minimum object resolution, with a maximum working distance of 50mm and an excitation illumination power of 52mW with 75° average field of illumination (meaning minimum device tip distance from target is 13mm for a 2cm polyp). The system demonstrated ex-vivo and in-vivo NIR visualization of clinically relevant concentrations of ICG in both resected and in situ (extracorporeally) colon in patients undergoing colorectal resection. A previously developed AI-NIR perfusion quantification algorithm was applied to videos of a bench model of varying ICG flow captured with the developed flexible system with added ML features generated full field of view pixel-level fluorescence time-series measurements capable of distinguishing distinct ICG flow regions in the image via correlative dynamic fluorescence intensity profiles. Jaccard Index comparison of the AI-generated flow regions against manually delineated flow regions resulted in 79% accuracy. While further clinical validation of the AI-NIR polyp classification method is on-going (in the Horizon Europe Awarded CLASSICA project), other use case applications of NIR colonoscopy include simpler perioperative perfusion assessment in patients undergoing colorectal resection and combination with targeted agents in development thus encouraging continuing development and design optimization of this flexible NIR imaging probe to enable clinical and commercial translation.

**Data availability statement:** All relevant data are within the paper.

**Funding:** Disruptive Technologies and Innovation Fund via Enterprise Ireland received by RC and PMcA. Url for Enterprise Ireland: https://www.enterprise-ireland.com/en/. Sponsors/funders played NO role in study design, data collection and analysis, decision to publish or preparation of manuscript. The funders had no role in study design, data collection and analysis, decision to publish, or preparation of the manuscript.

**Competing interests:** Professor Ronan A Cahill is named on patents awarded (Process for Visual Determination of Tissue Biology, PCT/EP2018/079359) and filed (Flexible Endoscopic NIR (near-infrared) Imaging Probe, PCT/EP 2023/063671 as well as Visual Perfusion Computation, PCT/EP 2022/076712; ICG Expert Interpreter, PCT/EP 2403605.5) in relation to this work and processes for visual determination of tissue biology including AI algorithmic methods. He also receives speaker fees from Stryker Corp and Ethicon/J&J, consultancy fees from Arthrex, Astellas, Diagnostic Green and Touch Surgery (Medtronic), research funding from Intuitive Corp and Medtronic as well as recently from the Irish Government (DTIF) in collaboration with IBM Research in Ireland, and from EU Horizon 2020 in collaboration with Palliare and, currently, from Horizon Europe in collaboration with Arctur. He is a member of the medical advisory board of Palliare.

# Introduction

Colorectal cancer (CRC) is the second leading cause of visceral cancer-mortality worldwide. Effective screening often utilizes direct intraluminal white-light visualization of the colon using a flexible imaging device. Such 'colonoscopy' has become broadly available and usefully impacts CRC incidence by enabling direct detection [1,2,3] and even some treatment of cancerous and precancerous lesions[4]. However, it has limitations including costs and complications and also both variability in small lesion detection rates and imperfect lesion characterization capability during the test especially for significant polyps (i.e., those > 2 cm in size) [5].

Along with improved bowel preparation and colonoscope optical performance [6], novel imaging technologies including narrow-band imaging (NBI) [7,8] and more recently artificial intelligence (AI) enabled computer-aided detection (CADe) systems [9,10] have improved small lesion detection rates[7,9,11]. Imperfect characterization however remains an important problem. This is because that, while small lesions can be safely removed upon colonoscopic detection, larger lesions may contain cancer and need different address [12] Local endoscopic excision (whether by trans-anal minimally invasive surgery, TAMIS, or endoscopic submucosal dissection, ESD) can cure well selected large benign and early-stage cancer lesions but adverse risk cancers need surgical oncologic resection[13] meaning accurate pre-excision discrimination is paramount. Misidentifying a cancerous lesion as benign and following the local excision treatment protocol may compromise subsequent treatment and worsen patient outcome[14]. Conversely, a benign lesion misdiagnosed as cancerous subjects the patient to unnecessary surgical resection with its associated higher morbidity, mortality and cost. At present, the clinical standard of endoscopist and biopsy categorization demonstrates at least 20% error rate in determining cancer presence in significant colorectal polyps due to cellular heterogeneity [15,16]. Furthermore biopsy sampling itself may induce local fibrosis [17] (potentially impacting any subsequent endoscopic resection) and the result is anyway only available sometime after the procedure. Therein lies a genuine need for better clinical care.

Near-infrared (NIR) fluorescence imaging may provide a suitable and sensitive tool to extract such information in real-time during evaluation. Its similar application in surgery for perfusion visualization is already widespread [18,19,20]. While autofluorescence and snap shot, point in time fluorescence imaging hours after agent administration have garnered mixed results re adenoma/adenocarcinoma distinction [21,22], recent work has shown efficacy with dynamic NIR fluorescence intensity timeseries analysis using the clinically approved fluorophore Indocyanine Green (ICG) and AI machine learning classification modelling [23,24,25]. ICG, lipoprotein bound in blood, demonstrates optical absorption ~ 780-805 nm and peak fluorescence ~ 820-830 nm and so avoids auto-fluorescent background noise [26,27, 28]. While continued development and clinical validation of the AI tissue classification model is ongoing via a multi-country implementation study (CLASSICA Horizon Europe [29]), the rigid nature of available fluorescence imaging systems (commercially available only as laparoscopes) limits the anatomical reach and potential of the classification technology. Commercially available flexible endoscopes' imaging capabilities do not yet extend to the NIR bandwidths. While researchers have explored multispectral flexible scopes [30,31,32,33,34], commercial translation has been hindered it seems by limitations in camera sensor hardware, optical properties of fiber optics and use of unregulated fluorophore agents. Provision of a flexible imaging platform that both visualizes ICG and accommodates implementation of AI algorithmic analysis of the imagery in a practical way could bridge the current gap. Provision to clinicians of ancillary accurate and easily actionable information to support treatment decisions could acuminate patient care and even potentially obviate traditional histopathology. An optically optimized device could also lend itself to additional fluorescent applications such as

tissue margination and intraluminal anastomosis assessments, along with other applications in other organs.

The purpose of this work was to manifest a useable and useful flexible NIR-ICG imaging probe with integrated AI-driven tissue classification methodology suitable for significant colorectal polyp evaluation throughout the colorectum. The device needs to be clinically translatable, and seamlessly fit current colonoscopy workflow. The optical performance of the developed device is reported here along with demonstration of its integration with AI fluorescence analysis both ex-vitro and ex vivo and in-vivo (extracorporeal use) during actual colorectal surgery in the clinical setting.

## Materials and methods

First the NIR-ICG imager was assembled to clinical design specification before proceeding to characterization testing and bench and clinical demonstration of proof of capability.

### NIR-ICG imaging system design

Initial investigational work optimizing the design concept and defining the input requirements has been reported previously [35] (Fig 1). To deliver on these specifications, an NIR-ICG imaging system was assembled comprising a multi-fiber optic bundle micro-endoscope, dual broadband and laser illumination sources, camera sensor with coupled wavelength filters and an image processor and display unit with each system unit combining to meet the flexible endoscopic fluorescence imaging format.

### Component specifications

The micro-endoscope (PD-PS-0095, Polyscope, Polydiagnost, Germany) incorporates two isolated fiber-optic bundles, one for imaging (10,000 pixel, FIGH-10-500N, Fujikura Ltd, Japan) and one for illumination. Together, these measure 2.4 mm in outer diameter across a 1.85 m working length, and thus are insertable via the working channel of conventional colonoscopes which are typically 3.7mm in diameter using a mother-daughter technique. Proximally, the micro-endoscope is tethered to a fixed-focus optical lens coupler (PD-FS-4001, Polydiagnost,

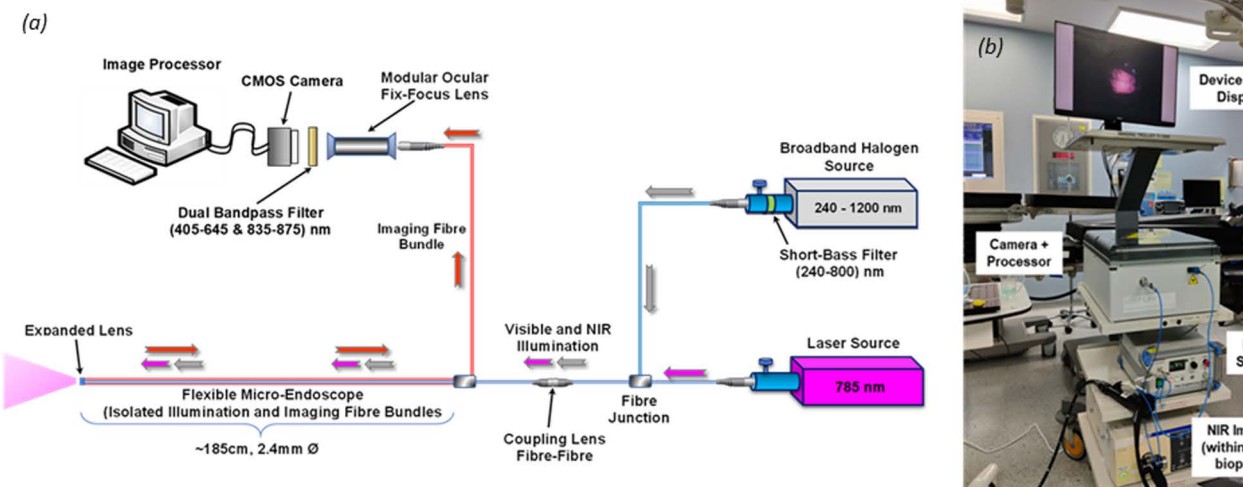

**Fig 1. Flexible Endoscopic NIR Imaging System (a); Built NIR Imaging System in Clinical Theatre Setting.**

Germany) compatible with many laparoscopic camera systems. The illumination light guide is coupled to a light adapter (PD-LC-9551, Polydiagnost, Germany), which adjusts expanded light to the fiber bundle. The more expensive imaging bundle can be easily removed from the outer catheter sheath making the device semi-disposable while reusing the imaging fiber optics as a glass lens at the catheter distal tip protects the imaging bundle from patient contact in situ. White light illumination and NIR excitation is derived from two independent sources: a broadband halogen bulb source (OSL2B2, Thorlabs, Germany) and a continuous 785 nm diode laser system (RLTMFC-785-1W-1, Roithner Lasertechnik, Austria), respectively. A y-bundle multimode fiber bundle (BF19Y2LS02, Thorlabs, Germany) transmits and collates both sources directing them through the micro-endoscope incorporated illumination guide via the light adapter. A 800 nm shortpass filter (FESH0800, Thorlabs, Germany), integrated between the broadband bulb and the y-bundle fiber adapter, removes potential reflectance noise that could cross contaminate the target visual field. Both source power intensities are adjustable such that the distal output can be tuned to meet the regulatory and safety requirements [36]. Distal light energy (both visible and NIR) is translated proximally along the fiber bundle length and relayed via the fixed-focus lens coupler through an additional 16 mm telescopic lens (SC0123, Raspberry Pi, UK) with integrated dual bandpass filter (useful ranges: 405-645 nm, 835-875 nm, DB850, Midwest Optical Systems Inc, USA). The filter removes laser excitation reflectance noise, while the lens focuses the collected image onto a single color complementary metal-oxide-semiconductor sensor (CMOS: IMX477R Sony, Japan). The camera (SC0261, HQ Camera, Raspberry Pi, UK), with 12 MP and 1.55x1.55 μm pixel size, was modified to remove the factory assembled infrared (IR) cutting filter in front of the 7.9 mm (diagonal) image sensor. The camera is connected and controlled directly from a 1.5 GHz 64 bit processor with 2 Gb RAM development board (Raspberry Pi 4 B, UK). Image acquisition is manually controlled and configured using the open source camera stack and framework, '*libcamera*' [37].

**SYSTEM ASSEMBLY** The system's optical components are mounted on a breadboard (MB3045/M, Thorlabs, Germany) and housed in a plastic enclosure (GEOS-L 4050-22-to, Spelsberg, UK). External polyscope connection interfaces were built into the front face of the enclosure utilizing mechanical supports and adapters where necessary, to ensure secure and stable fittings. The full system, including illumination sources, display unit (P2219H, Dell, USA) and device enclosure are mounted onto a medical cart (TI-1900, KeyMed, UK) for easy transport around the hospital and clinical environment.

## Optical performance characterization

**Illumination Power** (both for visualization and fluorescence excitation*)* was characterized by measuring the optical power at a 1 cm distance from the distal tip of the micro-endoscope at maximum illumination source setting (using PM100D, Thorlabs, Germany). Both broadband and laser sources were measured separately and the energy attenuation was determined across the length of the system.

**Bundle Angular field of illumination** (AFOI) was characterized by measuring the laser spotlight diameter (SD) at increasing working distances (WD) and calculating AFOI using the following formula (bundle optical angular field of view is 120º as specified by the manufacturer).

$$AFOI = 2tan^{-1}\left(\frac{SD}{2WD}\right)$$

**Resolution** The resolution of the system is limited by that of the imaging fiber bundle and needed to be determined by experiment. To do this, a standard 1951 United States Air Force

(USAF) resolution test target (R1DS1P, Thorlabs, Germany) [38] was employed and images of the three-bar group/element patterns were captured using white light illumination 3 mm from the target. The contrast transfer function (CTF) for each group/element in the captured image was determined by measuring the cross-sectional maximum ($I_{max}$) and minimum ($I_{min}$) greyscale values of the line/pairs and employing the Michelson contrast formula [39].

$$CTF = \frac{I_{max} - I_{min}}{I_{max} + I_{min}}$$

Plotting CTF against spatial frequency for both the horizontal and vertical patterns, with Rayleigh criterion [40,41,42] application, revealed the minimum observable object size indicate the true resolution.

**Sensitivity** for NIR-ICG fluorescence imaging was characterized by determining the sensing thresholds of the single color-sensor camera under expected clinical parameters. Recommended ICG concentration dosages range from 0.1-0.5 mg/kg body weight [43] depending on the application and anatomical target. As 0.25mg/kg was consistently used in the development and validation of the AI classification model [23,24,28] this was also used here. For testing, an ICG phantom was prepared using a 4% weight per volume bovine serum albumin (BSA, A3059, Sigma Aldrich, US) in a phosphate buffered saline (PBS, BR0014G, Fisher Scientific, US) as previously described [35], rendering a 4.6 µM ICG concentration. 0.4 mL of this solution. This was aliquoted into a single well (20 mm diameter x 5 mm depth) of a 3d printed (Ultimaker 3, Netherlands) 6x6 well plate made of black polylactic acid for with matt low reflectivity (the 20mm well diameter was chosen to simulate a 20mm polyp). A fresh ICG phantom was prepared prior to each test with a prepared BSA/PBS solution of the same volume also aliquoted into a separate well as a control measure. A fixed-exposure still image of each centered test well was captured. Each test measured average pixel signal intensity of a defined region of interest (ROI) located within the well area for the given ICG concentration. Multiple sensitivity tests were performed to determine the minimum threshold for the varied distance from target (well), illumination power intensity and camera gain control. The camera exposure setting was fixed for each test along with all other non-test parameters (i.e., distance from target and camera gain are fixed when testing illumination power). The signal to noise ratio (SNR) for each fluorescence test was calculated by formula where $S$ defines the average fluorescent pixel intensity in the ROI, $C$ is the average pixel intensity in the ROI of the BSA/PBS control image and $\sigma_C$ is the standard deviation of the pixel intensity from the control image ROI:

$$SNR = \frac{S - C}{\sigma_C}$$

## Dynamic perfusion model assessment

In order to integrate the fluorescence tracker software [28] and test the imaging system's ability to quantify dynamic fluorescence over time, a reproducible ex-vitro model was developed with the aim of simulating tissue perfusion inflow (wash in) and outflow (wash out) characteristics. The perfusion model was designed by building a fluid flow-circuit whereby a circulating pump (SC2718XPW DC 3V, Skoocom, China) infuses ICG via clear polyvinyl-chloride (PVC) tubing, towards a target of "mechanical-tissue" on which the NIR scope is focused. The administered ICG solution passes through the target chambers, subsequently disposing into a waste reservoir (Fig 2). From a physiological point of view, the rapid uptrend in fluorescence intensity arises from the first arrival of the intravenously administered ICG to

the targeted site. The subsequent intensity exponential decay constitutes the slow excretion of the exogenous fluorophore by the liver [44]. The variation in ICG intensity profile due to the differing vascular architecture of the underlying tissue (healthy, benign, malignant) serves as the marker for classification. Therefore, "mechanical-tissue" in the form of 3D printed (Form 3, Formlabs, USA) flow restrictors, using a clear stereolithography resin, encapsulated in transparent PVC tubing were designed, whereby the inlet and exhaust outlet orifices (0.8 mm and 1.2 mm diameter outlets) vary the flow rate between each chamber. ICG enters the target chambers via the restrictor inlet, subsequently flowing out and filling the space between the restrictors and the clear PVC tubing and finally escapes the chamber via the restrictor outlet exhaust orifice. The NIR-ICG imaging system captured the varied ICG flow rates through the chambers that ultimately produce differing intensity profiles for quantification by the fluorescence tracking software.

## AI-assisted perfusion margination assessment

The fluorescence intensity tracker-quantifier software first proposed in [28] was developed and tailored for use with specific fluorescent and white light clinical videos captured from a clinically approved fluorescence imaging stack (Pinpoint Endolaparoscopic Imaging System, Stryker, USA) during surgical operations. The aim of this was to quantify and distinguish fluorescence perfusion patterns in selected target tissue ROIs by comparing inflow and outflow behaviors/features. In its current configuration, the fluorescence intensity tracker-quantifier requires pre-selection of ROI for assessment which is inherently prescriptive and requires the user to possess some level of experience when assessing a NIR image of colorectal tissue, where variability in user interpretation is evident [45]. Ideally a full field of view (FOV) assessment with user feedback in the form of boundary delineation of tissue types via differing perfusion patterns is the desired end product functionality. A python-based machine learning algorithm (capable of pixel level fluorescence measurement, clustering of similar intensity time-series and mapping of clusters) was developed to satisfy this need. Fluorescent videos captured from the imaging system and the aforementioned perfusion model, provided the input data for the algorithm development with the end goal of it distinguishing each perfusion flow chamber.

The algorithm structure is comprised of three main operations (1) input video pre-processing, (2) machine learning clustering (grouping similar time-series pixels) and (3) similarity assessment via correlation. The input color video is primed by isolating the most fluorescent sensitive RGB channel and filtering out pixels that do not surpass an intensity threshold, discarding image background and areas of non-dynamic fluorescence. Principal component analysis (PCA) was implemented to reduce the overall model complexity. Clustering of these approximated pixel time-series should reveal pixels associated with each ICG

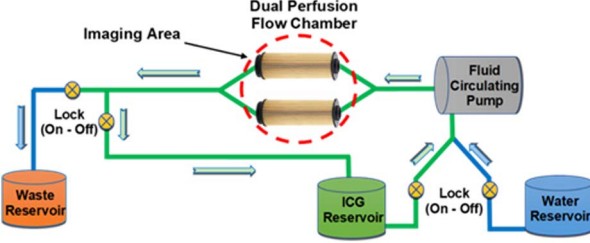

**Fig 2. Perfusion model schematic.**

flow chamber. Finally, the clusters most representative of chambers (i.e., those with highest accumulated pixel intensity), are selected and each pixel signal is tested for similarity to either cluster signal using normalized cross correlation (NCC). For two signals $a(t)$ and $b(t)$ of equal lengths $T$, where $\bar{a}$, $\bar{b}$ are the averages of $a$ and $b$, and $s_a$, $s_b$ are the sample standard deviations, the NCC is defined as:

$$NCC = \frac{1}{T-1}\left(\sum_{t=1}^{T}\frac{a(t)-\bar{a}}{s_a}\frac{b(t)-\bar{b}}{s_b}\right)$$

If $a(t)$ and $b(t)$ are exactly the same, NCC should be 1, if $a(t) = -b(t)$ it is -1, and in general, the closer to 0, the less similar the signals are. A pixel level NCC for each cluster center time-series ($NCC_{C1}$, $NCC_{C2}$) is generated and masked onto the perfusion model image, indicating a correlation gradient. Subtracting the NCC clusters reveals an approximate delineation of each of the flow chambers ($NCC_{Diff}$). The end output is an algorithmically generated heatmap of pixels illustrating their similarity to the overall fluorescence curves of either flow chamber.

To assess the accuracy of the AI model in delineating the area of each flow chamber in a given image, the Jaccard Index (J) [46], a statistical measurement of similarity between two data sets (A, B), was employed. For each AI generated $NCC_{Diff}$, binary matrices (A), of size 640x480 (image resolution) were exported for each flow chamber representation, whereby all the pixels indicated from the $NCC_{Diff}$ for a given chamber are given a value of 1, and all remaining pixels are given a value of 0. For comparison, individual flow chamber binary matrices (B) were generated using ImageJ (National Institutes of Health, US) by manually drawing the outline of each chamber from a given NIR scope captured image and scoring the pixels within the boundary 1 and all others 0. The percentage similarity of the AI generated versus the manually drawn flow chamber boundary for each individual chamber for a total of N = 9 NIR flexible scope captured fluorescent videos was determined thus:

$$J(A,B) = \frac{|A \cap B|}{|A \cup B|} = \frac{|A \cap B|}{|A| + |B| - |A \cap B|}$$

## Clinical testing

As part of an Institutional Review Board approved clinical trial (IRB approval 1/378/2092, ClinicalTrials.gov Identifier: NCT04220242), the NIR-ICG imaging device was clinically tested ex-vivo and in-vivo in order to demonstrate the systems clinical feasibility and to assess its qualitative performance. Participants provided written informed consent for inclusion in this study with patients for this aspect being recruited between January and June 2023.

## Ex-vivo assessment- human specimen imaging

During a colorectal cancer patient surgical procedure, 0.25 mg/kg ICG was administered intravenously as routinely done to assess bowel perfusion during laparoscopic resection of the descending colon. Post-resection the bowel specimen was transferred to a prep room adjacent to the theatre to allow NIR-ICG imaging system testing. Fluorescence images and videos were recorded of the specimen and remaining ICG absorbed by the tissue, first, with freehand control of the probe, and subsequently in a controlled perpendicular fashion at both fixed and varied distances and camera gains settings. The excitation power of the system remained constant at 30 mW.

### In-vivo assessment- patient stoma imaging

The NIR imaging system was assessed for usability and qualitative performance in a clinical theatre setting during two separate rectal cancer resections. Under the control of the surgical consultant, the system was used extracorporeally without patient contact, to image the dynamic inflow of ICG extracorporeally at the site of the patient stoma. 0.25 mg/kg ICG was administered intravenously, while the system employed a 50 mW excitation energy. A full-length video of the stoma tissue under observation and resultant fluorescence profile with the surgeon moving the probe tip was recorded.

## Results

### Optical performance characterization (Table 1)

Flexible NIR Imaging System Performance Specifications are shown in Table 1

### Illumination power

The system produced a maximum 52 mW laser excitation illumination 10mm from the distal probe tip. This constitutes a 12.84 dB attenuation across the probe length. The maximum distal broadband illumination was measured at 3mW at a 10mm distance with 47 dB attenuation across the probe length. With respect to the laser safety standard, IEC 60825 [36], the combined energy output of the system is within the accessible emission limit (AEL) (approx. 100mW for a device of this configuration) with 3R classification.

**Table 1. NIR Imaging System Specification.**

| Physical Specification | Imaging Probe Length | 1.85m | |
|---|---|---|---|
| | Probe Outer Diameter | 2.4mm | |
| | Working Channel Inner Diameter | 1.2mm | |
| | Device Enclosure Size (LxWxH) | 400x500x226mm | |
| Illumination/ Excitation | Light Source Type | Broadband Halogen Bulb | Laser Diode |
| | Wavelength (nm) | 400 - 1750nm | 785nm |
| | Operation Mode | Continuous | Continuous |
| | Wavelength Filter | 800nm shortpass | N/A |
| | Fiber Beam Divergence (AFOV) | 75° | 75° |
| | Power Output (max) | 150W | 1W |
| | Power Output 10mm distance to Probe Tip (max) | 3mW | 52mW |
| Imaging | Transmission Spectrum (nm) | 405-645nm, 835-875nm | |
| | Sensor Type | CMOS | |
| | Sensor Size | Diagonal 7.857mm (Type 1/2.3) | |
| | Pixels | 4056x3040 | |
| | Pixel Size | 1.55x1.55μm | |
| | Video Framerate | 30fps | |
| | Video Resolution | 640x480 | |
| Optics | Fiber Resolution | 10,000 Pixels | |
| | Fiber Image Circle Diameter | 460μm | |
| | Working Distance (cm) | 15-50mm | |
| | Angular Field of View (AFOV) | 120° | |

## Illumination field of view

Incremental distances ranging from 5-20 mm were tested and the laser spotlight diameter measured. The resultant average AFOI was calculated to be 75° (i.e., less than the full system image field of view). In order to capture and uniformly excite a 2 cm polyp, the minimum device probe tip distance from the target, based on the 75° AFOI, is approximately 13 mm. Therefore, the minimum working distance of our system is recommended to be 15 mm but is greater for targets (i.e.,) polyps greater than 2 cm diameter.

## Resolution

Employing the USAF 1951 pattern target and the Michelson contrast formula helped determine the true spatial resolution of the developed fluorescence imaging system (Fig 3). Greyscale intensity values were measured for each of the group/element line-pair patterns discernable from the captured image of the test target, while the CTF was calculated from the respective maximum and minimum intensities. Applying the Rayleigh criterion, in which the contrast limit between two objects is approximately 26.4%, reveals a minimum horizontal and vertical spatial resolution of 3.763 and 3.24 line pair/mm. Taking the half-width between the two line pairs the corresponding to a minimum transverse and lateral observable object was determined to be 133 μm ($R^2$ is 0.886) horizontal and 154 μm ($R^2$ is 0.9072), respectively.

## Sensitivity

The SNR was plotted against each test parameter below (Fig 4). The distance testing was performed with a 50 mW laser power excitation and a camera gain setting of 50. The fit of the plotted results decreases to the power of -1.442 with increase in distance, consistent with the inverse square law. Based on the results the excitation power testing demonstrated a linear relationship, as expected, while so too was the relationship between the SNR and camera gain. Based on the results, the recommended maximum working distance is 50 mm from the target (signal approximately 10 times the noise), with a gain of 50. The maximum achievable laser power excitation of approximately 50 mW maximizes the available fluorescence potential, while still falling 50% below the allowable safety limit for a laser of this kind (class 3R from IEC 60825-1 [36]).

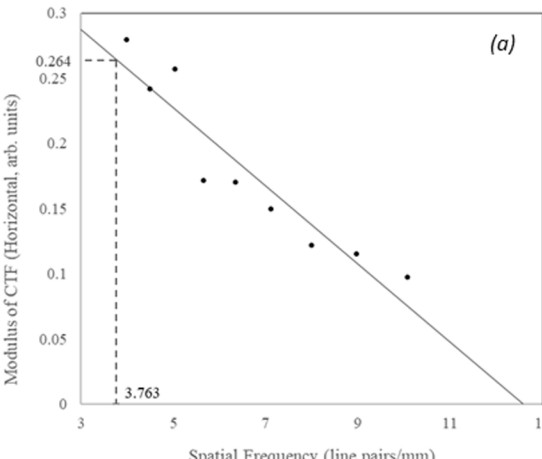 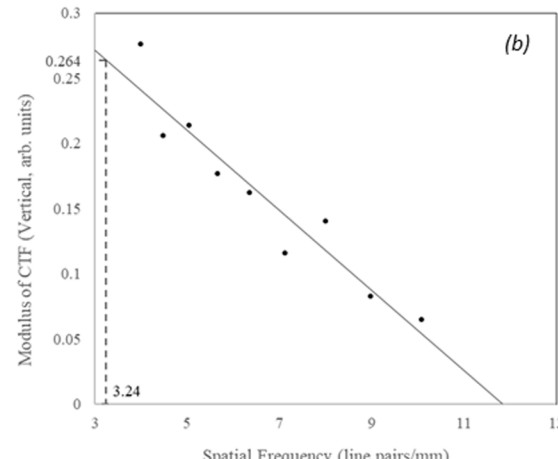

**Fig 3. Horizontal (a) and Vertical (b) contrast transfer function from each tested bar/group patterns captured from the USAF 1951 resolution test target image.**

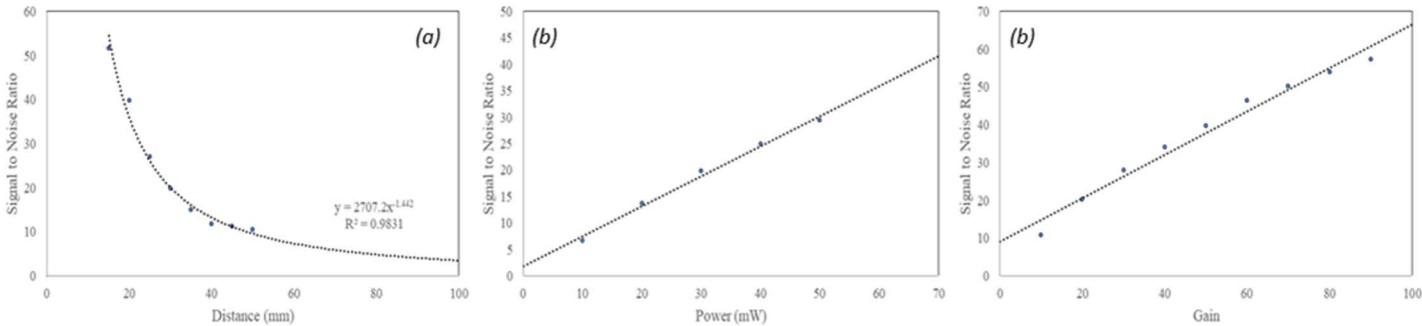

**Fig 4. System sensitivity SNR relation to distance (a), excitation power (b) and camera gain setting (c).**

### AI-assisted perfusion margination assessment

The developed AI driven perfusion quantification algorithm was qualitatively successful in flow chamber identification and margination (Fig 5). The heatmap plot (Fig 5(h)) depicting the $NCC_{Diff}$, effectively displays the flow chamber area within the image resolution, while also differentiating the two. The NCC for each cluster shows high correlation ($NCC \approx 1$) against all pixels in the perfusion image area, indicating minor quantitative variations in perfusion across each chamber. This further highlights the power of the quantitative approach and algorithms capability of detecting subtle differences in the perfusion pattern.

A total of nine dynamic perfusion videos were captured by the NIR-ICG system and used to assess the similarity of the algorithmically generated heatmaps in defining the flow chamber boundary compared to a human-manually outlined chamber. While maintaining each flow chamber within the field of view, the image perspective and imager distance to the flow chambers for each captured video varied. The Jaccard Index provided a means to compare the AI generated chamber to the human generated for a given video (Fig 6). The average Jaccard similarity score was 79% (62% min, 94% max) across all video samples and flow chambers. Although this is a fair comparison, the inherent inaccuracies of a manually outlined flow chamber do not result in a true test of the algorithms effectiveness in delineating each flow chamber. As observed with the high correlation between chambers generated from the NCC, it is likely the quantification capabilities of the algorithm could well indeed provide a more accurate and specific chamber outline based off ICG intensity profiles compared to a human's free hand outline and context of each flow chamber within a given image.

### Clinical testing

The developed flexible NIR system visualized ICG fluorescence of typical concentrations in a clinical surgical setting in both ex vivo and in vivo trialing (Fig 7).

### Discussion

Given the existing and increasing application of NIR visualization in minimally invasive and indeed open surgical practice, the evolution of flexible endoscopy to include fluorescence and multispectral imaging would seem to be an inevitable increment in its technological and functional development. As the optimal management of significant colorectal polyps presents a difficult balance between treatment strategy, patient risk and procedural cost [47,48,49,50,51], especially given the inaccuracies associated with biopsy of larger morphologies [14], this would seem to be an ideal use case to advance this. However additional potential uses including the intraoperative and postoperative assessment of colonic perfusion, a key determinant

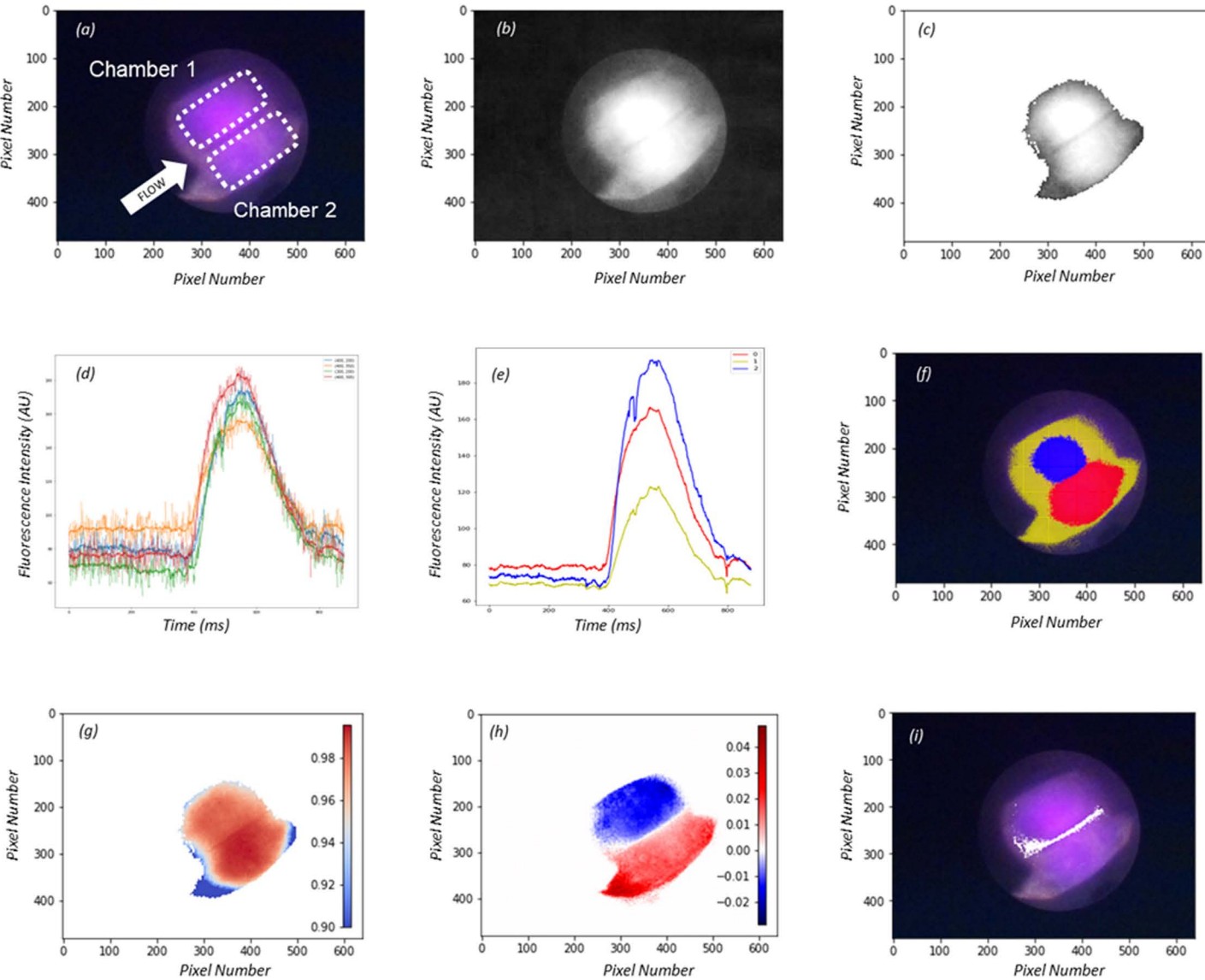

**Fig 5. AI Algorithm Workflow: Image pre-processing with highest pixel intensity identification (a), RGB channel breakdown, Fluorescence isolation and grayscale conversion (b), Computational reduction via pixel intensity thresholding (c), Principal component analysis (PCA) implementation for time-series dimensionality reduction (d), K-means clustering of reduced dimensional vector to reveal similar time-series (e), Visual representation of clustering on flow chamber image, chamber-representative cluster selection is based on accumulated fluorescence (area under curve) (f), Normalized cross correlation (NCC) of pixel time-series with cluster 1.** Red indicates high correlation (NCC ≈ 1) (g), NCC cluster differentiation reveals perfusion flow chambers (NCC1-NCC2 > 0, red; NCC1-NCC2 < 0, blue) (h), Visual representation of chamber margin (white) on image (NCC1-NCC2 ≈ 0) (i).

of anastomotic healing and driver of management strategy in event of anastomotic breakdown, as well as cancer identification in combination with novel targeted fluorophores now in clinical trials. Importantly, AI methods are recently too being applied to ICG perfusion use with clinical validation indicating promise [52,53,54]. Of course such a probe could also add value in other forms of endoscopy including upper gastrointestinal tract, respiratory and ENT endoscopy for similar physiological and pathological indications.

Focusing here on the use case of significant colorectal polyp in situ characterization using ICG and AI methods, the primary objective for developing the system presented in this work was to extend effective NIR visualization capability to areas of the colon beyond where the

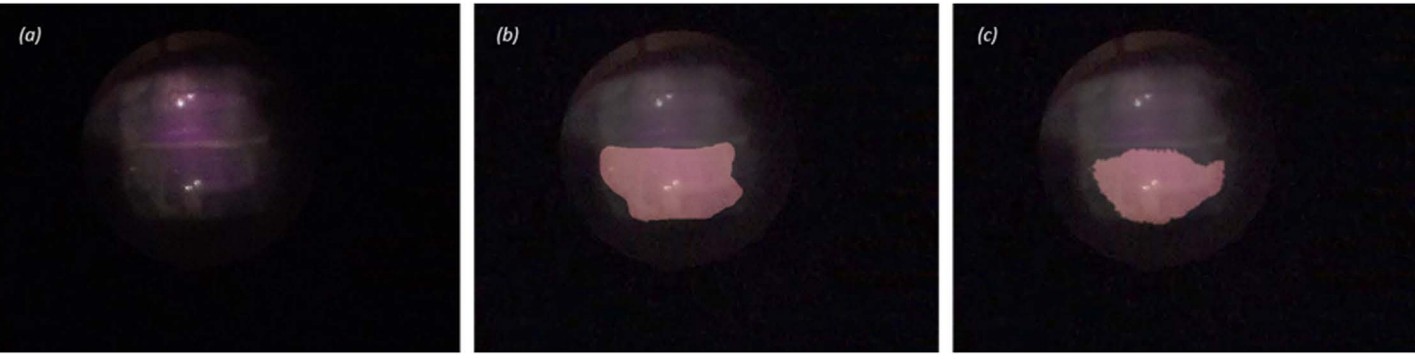

**Fig 6. Sample dual perfusion chamber (top and bottom) image (a), manually generated bottom chamber outline overlayed on original image (b), AI generated bottom chamber outline overlayed on original image (c).**

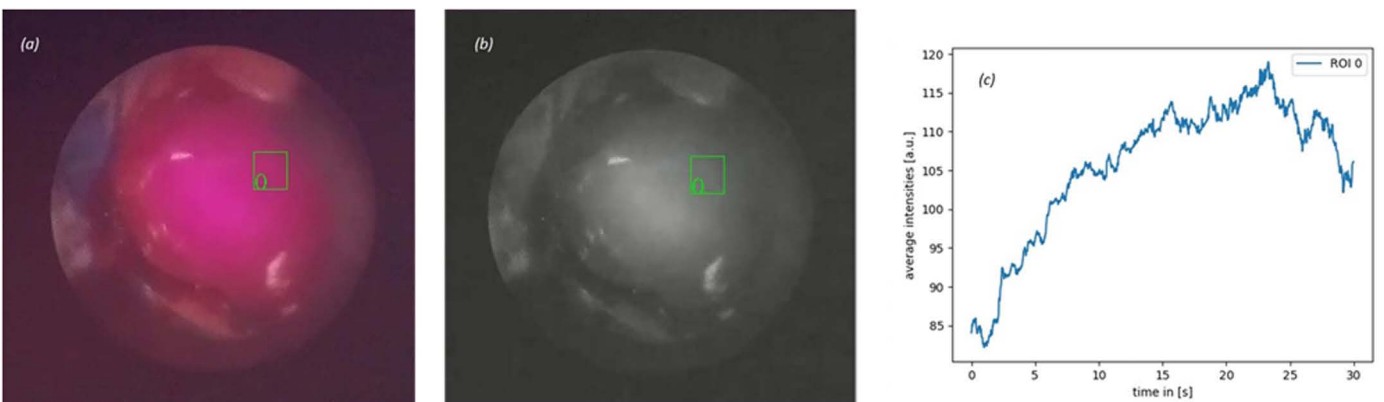

**Fig 7. Fluorescence image of patient stoma with pre-selected ROI taken from video captured by the NIR system during surgery (a), greyscale conversion of captured image (b), fluorescence intensity tracking of ROI (c).**

current commercial NIR imaging systems can reach (i.e., the colon proximal to the rectum). Subsequently the aim was to demonstrate the capacity for real-time AI-assisted tissue classification via ICG fluorescence quantification as the key consideration for clinicians after identification of significant polyps is determination of lesion nature as this drives subsequent care. This device so aims to replace conventional biopsy (and indeed it occupies the same working channel as a biopsy forceps would use) by providing actionable information for the clinician to proceed either immediately to resection, refer to another practitioner or center with greater endoscopic expertise or capability or, alternatively, proceed down the cancer care pathway (biopsy and radiological staging before multidisciplinary treatment planning). Currently no commercial device provides such technical or technological capability with systems only providing polyp identification or methods that require considerable user experience and interpretation of presented augmented visual appearances (whether by specific light enhancement or indeed microscopic image presentation, e.g., Cellvizio, Mauna Kea Technologies, France). None of these therefore supply the automatic in situ characterization needed to really optimize colonoscopic encounter with significant colorectal lesions at scale (colonoscopies are performed most widely in secondary healthcare centers).

The work and system presented here developmentally meets these needs via the manifestation of a colonoscope compatible flexible imaging probe for NIR visualization combined

with a prototype full field of view machine learning algorithm for fluorescence quantification and perfusion pattern cross correlation. Usability of the device is indicated from both in and ex-vivo testing and the fact that it integrates into colonoscopy workflow without disrupting current technology nor, in the first instance, making histopathology redundant are strengths of the method. While in the oesphagus and stomach, squamous cell cancers become more common these are also still likely to be characterizable in a similar way to polyps, as the system characterizes microperfusion rather than specific tissue architecture and the small caliber of the tool also means potential for use in characterization of lesions of the biliary tree (whether by endoscopic or laparoscopic access) and indeed urological and gynecological surfaces. Furthermore, while the system presented here is predicated on ICG as the only currently approved and widely available fluorophore, other additional NIR agents in development for both cancer, anatomic structure, and physiology/pathology discrimination would likely benefit from capable visualization with some augmented, objective quantification (to minimize issues with background fluorescence or false positive artefacts) to ensure their best use by the widest user base to ensure maximum patient benefit.

Alongside further validation of the AI tissue classification (this is currently progressing in a Horizon Europe Awarded CLASSICA project [29]), continued development and design optimization of the flexible NIR imaging probe can complete the necessary next steps for clinical and commercial translation. In particular, the optical performance of the device needs to improve with regard to clinically relevant parameters such as widening the field of view (its present functional requirement in this regard may pose limitations) and providing improved sensitivity at larger working distances. An optically ideal system would mimic performance of a HD wide-angled colonoscope. Next steps so will include consideration of chip-on-tip solutions along with specific wavelength sensors in this regard. With regards to the existing AI characterization software, immediate next steps are graphical user interface (GUI) development and more integrated AI-endoscopic image display with software to control and process image acquisition (likely using python and opencv tooling) with control unit miniaturization. Advancing the AI-algorithm to function as a margination algorithm requires of course full field of view assessment but also could make the optical presentation more efficient by filtering out pixels not under fluorescence. However, there will still be a processing time required and it seems unlikely that margination can be integrated onto the current system processor and so may represent a second intraluminal application.

In summary, the developed system in this work enables a step advance in contemporary and future flexible endoscopic clinical practice by extending NIR visualization capability to more torturous areas of the gastrointestinal tract while preserving current clinical practice workflows and existing equipment and infrastructure. It also implements computer vision and AI methods for real-time fluorescence imaging and subsequent classification of tissue in a method that is clinically appealing and that can already leverage an existing, approved medical agent.

## Acknowledgements

The Authors posthumously acknowledge the contribution of Prof. John Sheridan of University College who would otherwise be listed as a co-author on this work for his invaluable guidance and input.

## Author contributions

**Conceptualization:** Ronan A Cahill.

**Data curation:** Ronan A Cahill, Ra'ed Malallah.

**Formal analysis:** Gareth Gallagher, Ra'ed Malallah, Jeffrey Dalli, Niall Hardy, Abhinav Jindal, Pol G Macaonghusa.

**Funding acquisition:** Ronan A Cahill, Pol G Macaonghusa.

**Investigation:** Ronan A Cahill, Gareth Gallagher.

**Methodology:** Ronan A Cahill, Gareth Gallagher, Ra'ed Malallah.

**Project administration:** Ronan A Cahill, Gareth Gallagher.

**Resources:** Ronan A Cahill.

**Supervision:** Ronan A Cahill, Pol G Macaonghusa.

**Validation:** Gareth Gallagher, Ra'ed Malallah, Jonathan P Epperlein, Jeffrey Dalli.

**Writing – original draft:** Ronan A Cahill, Gareth Gallagher.

**Writing – review & editing:** Ronan A Cahill, Ra'ed Malallah.

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
