## [Decision Letter · Decision Letter 0]

11 Dec 2024

PONE-D-24-13285A Novel Flexible NearInfrared Endoscopic Device that Enables Realtime Artificial Intelligence Fluorescence Tissue CharacterizationPLOS ONE

Dear Dr. Cahill,

Thank you for submitting your manuscript to PLOS ONE. After careful consideration, we feel that it has merit but does not fully meet PLOS ONE’s publication criteria as it currently stands. Therefore, we invite you to submit a revised version of the manuscript that addresses the points raised during the review process.

We look forward to receiving your revised manuscript.

Kind regards,

Diego Raimondo

Academic Editor

PLOS ONE

Journal Requirements:

2. Please note that PLOS ONE has specific guidelines on code sharing for submissions in which author-generated code underpins the findings in the manuscript. In these cases, all author-generated code must be made available without restrictions upon publication of the work. 

Please review our guidelines at https://journals.plos.org/plosone/s/materials-and-software-sharing#loc-sharing-code and ensure that your code is shared in a way that follows best practice and facilitates reproducibility and reuse.

3. Please note that funding information should not appear in the Acknowledgments section or other areas of your manuscript. We will only publish funding information present in the Funding Statement section of the online submission form. Please remove any funding-related text from the manuscript. 

4. We note that you have a patent relating to material pertinent to this article:

“Professor Ronan A Cahill is named on patents filed in relation to processes for visual determination of tissue biology including AI algorithmic methods and receives speaker fees from Stryker Corp and Ethicon/J&J, consultancy fees from Arthrex, Astellas, Diagnostic Green and Touch Surgery (Medtronic), research funding from Intuitive Corp and Medtronic as well as recently from the Irish Government (DTIF) in collaboration with IBM Research in Ireland, and from EU Horizon 2020 in collaboration with Palliare and, currently, from Horizon Europe in collaboration with Arctur.”

Please provide an amended statement of Competing Interests to declare this patent (with details including name and number), along with any other relevant declarations relating to employment, consultancy, patents, products in development or modified products etc. Please confirm that this does not alter your adherence to all PLOS ONE policies on sharing data and materials, as detailed online in our guide for authors http://journals.plos.org/plosone/s/competing-interests by including the following statement: "This does not alter our adherence to  PLOS ONE policies on sharing data and materials.” If there are restrictions on sharing of data and/or materials, please state these. Please note that we cannot proceed with consideration of your article until this information has been declared.

5. We note that your Data Availability Statement is currently as follows:

“All relevant data are within the manuscript and its Supporting Information files.”

6. Please ensure that you refer to Figure 3 in your text as, if accepted, production will need this reference to link the reader to the figure.

7. Please upload a copy of Figure 6, to which you refer in your text on page 10. If the figure is no longer to be included as part of the submission please remove all reference to it within the text.

Reviewers' comments:

Reviewer's Responses to Questions

**Comments to the Author**

1. Is the manuscript technically sound, and do the data support the conclusions?

Reviewer #1: Partly

2. Has the statistical analysis been performed appropriately and rigorously? 

Reviewer #1: N/A

3. Have the authors made all data underlying the findings in their manuscript fully available?

Reviewer #1: Yes

4. Is the manuscript presented in an intelligible fashion and written in standard English?

Reviewer #1: Yes

5. Review Comments to the Author

Reviewer #1: Dear Authors,

I read you article presenting your new colonoscope-compatible flexible imaging probe for NIR visualization with its AI algorithm for ICG quantification.

It raised my interest as well as some concerns over its structure. I listed them as follows:

- the structure of the article is not clear in my mind. It is a research article but it seems more of a leaflet presenting a new instrument. Actually it is not an ex-vivo study investigating the application of the instrument, nor it is a clinical article. Hence, what's the aim of the study?

- overall, the flow of the article should be increased. There are too many parenthesis, making it difficult to read.

- Methods section should be deeply revised in its structure. Please describe the design of the study and insert subsections, more than you did (i.e., description of the instrument)

- It seems like the clinical testing section of the results is halved or has a missing part

- Please discuss the potential clinical application of AI for the detection of ICG in different settings, still evaluating the bowel (i.e., perfusion, PMID: 33352315)

6. PLOS authors have the option to publish the peer review history of their article (what does this mean? ). If published, this will include your full peer review and any attached files.

**Do you want your identity to be public for this peer review?** For information about this choice, including consent withdrawal, please see our Privacy Policy .

Reviewer #1: No

---

## [Author Response · Author response to Decision Letter 0]

20 Dec 2024

20th December 2024.

Dear Dr Raimondo and Editorial Office and Review team,

Thank you for the comprehensive review and comments regarding our manuscript. We are pleased to return here a revised version of the manuscript in clean and tracked version that address all points raised during the review process. We remain fully available to work further on the manuscript at your direction to make sure the best version of this work is presented for the journal’s audience and, indeed the field.

Re revision requirement 1-3: Firstly, we confirm that the revision now adheres to the journals style requirements specifically as advised with regard to headings, figure and table citations and positions and indeed reference formatting.

Re requirement 4: The funding acknowledgment has been moved out of the acknowledgment section and the patent numbers and titles referred to have now been added explicitly as requested in the Disclosures section.

Re requirement 5: We confirm that all raw data as need to replicate these experiments is included and detailed in the manuscript. This indeed is confirmed as per Reviewer 1 comments. A statement to this effect has been added to the manuscript itself as well for complete clarity.

Re requirement 6 and 7: Figure 3 this is now included as a citation in the text and we confirm too that all others are as well and we have uploaded a copy of Figure 6, thank you. Figures are now all supplied separately and in 300dpi format.

To confirm too re the role of the funder “The funders had no role in study design, data collection and analysis, decision to publish, or preparation of the manuscript.”

Regarding the reviewer specific comments please find here point by point commentary:

Comments to the Author 1. Is the manuscript technically sound, and do the data support the conclusions? The manuscript must describe a technically sound piece of scientific research with data that supports the conclusions. Experiments must have been conducted rigorously, with appropriate controls, replication, and sample sizes. The conclusions must be drawn appropriately based on the data presented. Reviewer #1: Partly

Author reply: We have carefully and explicitly delineated the construction via assembly of the device that is the subject of the report in such a way to allow others do so similarly and now supply a revised manuscript that includes the specific comments of the editorial office and indeed your own review. We agree this makes for a clearer, better structured report thank you! Please let us know if any further detail would be helpful to strengthen the report.

2. Has the statistical analysis been performed appropriately and rigorously? Reviewer #1: N/A

Author reply: We agree that statistical analysis is not a significant part of the meaning and message of the report, thank you.

3. Have the authors made all data underlying the findings in their manuscript fully available? The PLOS Data policy requires authors to make all data underlying the findings described in their manuscript fully available without restriction, with rare exception (please refer to the Data Availability Statement in the manuscript PDF file). The data should be provided as part of the manuscript or its supporting information, or deposited to a public repository. For example, in addition to summary statistics, the data points behind means, medians and variance measures should be available. If there are restrictions on publicly sharing data—e.g. participant privacy or use of data from a third party—those must be specified. Reviewer #1: Yes

Author reply: We agree!

4. Is the manuscript presented in an intelligible fashion and written in standard English? PLOS ONE does not copyedit accepted manuscripts, so the language in submitted articles must be clear, correct, and unambiguous. Any typographical or grammatical errors should be corrected at revision, so please note any specific errors here. Reviewer #1: Yes

Author reply: Thank you!

5. Review Comments to the Author. Please use the space provided to explain your answers to the questions above. You may also include additional comments for the author, including concerns about dual publication, research ethics, or publication ethics. (Please upload your review as an attachment if it exceeds 20,000 characters) Reviewer #1: Dear Authors, I read you article presenting your new colonoscope-compatible flexible imaging probe for NIR visualization with its AI algorithm for ICG quantification. It raised my interest as well as some concerns over its structure.

Author reply: Thank you for your careful review and comments, which we address now here below in order..

I listed them as follows:- the structure of the article is not clear in my mind. It is a research article but it seems more of a leaflet presenting a new instrument. Actually it is not an ex-vivo study investigating the application of the instrument, nor it is a clinical article. Hence, what's the aim of the study?

Author reply: The aim is to manifest a method that closes the current clinical gap in near infrared imaging articulated in the manuscript. That is that current NIR imagers aren’t flexible and flexible imagers don’t have NIR. We have revised the abstract and introduction to make this clearer. Thanks for the recommendation!

- overall, the flow of the article should be increased. There are too many parenthesis, making it difficult to read.

Author reply: we have revised the manuscript fully with this specific comment in mind and have, as you recommend, improve the flow for instance by making simpler, shorter sentences where appropriate. We agree this has greatly helped the flow of the paper making the message clearer and indeed the report shorter! Thank you!

- Methods section should be deeply revised in its structure. Please describe the design of the study and insert subsections, more than you did (i.e., description of the instrument)

Thank you, this has now been done. No problem to address further if you think could be improve further.

- It seems like the clinical testing section of the results is halved or has a missing part

This has been corrected, we agree it looked like a piece was missing but this has now been cleaned up.

- Please discuss the potential clinical application of AI for the detection of ICG in different settings, still evaluating the bowel (i.e., perfusion, PMID: 33352315)

This is now done with inclusion of your suggested reference and some others. Thank you!

---

## [Editor Report · Decision Letter 1]

5 Jan 2025

A Novel Flexible NearInfrared Endoscopic Device that Enables Realtime Artificial Intelligence Fluorescence Tissue Characterization

PONE-D-24-13285R1

Dear Dr. Cahill,

We’re pleased to inform you that your manuscript has been judged scientifically suitable for publication and will be formally accepted for publication once it meets all outstanding technical requirements.

Kind regards,

Diego Raimondo

Academic Editor

PLOS ONE

---

## [Editor Report · Acceptance letter]

PONE-D-24-13285R1

PLOS ONE

Dear Dr. Cahill,

I'm pleased to inform you that your manuscript has been deemed suitable for publication in PLOS ONE. Congratulations! Your manuscript is now being handed over to our production team.

Kind regards,

on behalf of

Dr. Diego Raimondo

Academic Editor

PLOS ONE